# Transformation of Paddy Field Use in Intermountain-Type Basins Using Evidence from the Structure and Function Perspective of Karst Mountain Areas in Southwest China

**Mei Chen [1], Yangbing Li [1,\*], Fang Tang [1]**  **, Qian Xu [2], Meng Yu [1], Han Zhang [1] and Xue Li [3]**

[1] School of Geography and Environmental Sciences, Guizhou Normal University, Huaxi District, Guiyang 550025, China; mui-chen@gznu.edu.cn (M.C.); 19010090220@gznu.edu.cn (F.T.); 20010090296@gznu.edu.cn (M.Y.); 19010090227@gznu.edu.cn (H.Z.)

[2] School of International Tourism and Culture, Guizhou Normal University, Huaxi District, Guiyang 550025, China; 15030090010@gznu.edu.cn

[3] Ecological Meteorology and Satellite Remote Sensing Center of Guizhou Province, Nanming District, Guiyang 550002, China; lixue1019@163.com

\* Correspondence: li-yabin@gznu.edu.cn; Tel.: +86-138-8514-1624

**Abstract:** China's southwestern karst area exhibits many mountains and little flat lands, and intermountain basins (IBs) (locally named "bazi") are one of their typical landform types. Comparative studies on land use in the core of typical landforms in karst mountain areas are relatively lacking. Studying the evolution and transformation patterns of intermountain basin paddy fields use (IBPFU) to optimize land resources in mountainous areas is important. Based on structural and functional perspectives, this study established a research framework on the evolution and transformation of IBPFU in karst mountainous areas, and analyzed the evolution and transformation patterns of IBPFU in Guizhou Province from 1966 to 2020 by measuring land use change and applying the moving window method and morphological spatial pattern analysis (MSPA) model. The study showed that the evolution of IBPFU is characterized by reversibility and irreversibility, diversity, and stages. The transition of IBPFU requires a trade-off among regional socioeconomic development, ecological protection, and food security. The commonality of land use transitions in IB and sloped land (SL) is reflected in the diversity of their land use functions. This study provides a reference for a comprehensive understanding of land use transitions and sustainable development in the mountainous karst regions of southwest China by providing in-depth examinations of the land use transition of IB, which is different from SL, via the long time series evolution of core land use types. The research framework and research method constructed here also apply to other regions.

**Keywords:** paddy fields; transformation; driving mechanisms; intermountain basins; southwest China

## 1. Introduction

As a critical cultivated land resource, paddy fields play an essential role in national and global food security [1,2], and their changes are closely related to human activities, such as in Asia and Africa [3], where urban expansion occupied substantial arable land and led to food system instability; moreover, in India [4], the government protected paddy fields resources by enacting policies. Spatially, their changes manifest themselves in the expansion or reduction in the area of paddy fields in a region [5,6], and they have an important impact on agricultural production functions and agro-ecosystem services in the region [7]. In recent years, driven by urbanization and industrialization, a large proportion of paddy fields have been converted to construction land [8–10] and dry land for cash crops [11], and the paddy landscape has become increasingly fragmented [12–14], affecting food security [15]. Studying the evolution and transition characteristics of paddy field use is vital for sustainable land resource management, paddy field protection policies, and food security [16,17].

Currently, research on paddy fields includes the impact of dry land to paddy conversion on food production [15]; the impact of farmland restoration on food and nature [18]; food security and the net carbon fixation value of paddy fields systems [19]; crop diversification [20]; ecological service value [21]; effects on soil and response to climate [22–24], etc. However, studies by scholars on the mechanisms driving the transformation of single-paddy land use in the region are relatively rare, and most are focused on land use or arable use in the region [25]. Research on paddy field use has a clear geographical distribution, and this has mainly been carried out in Asia, western and central Europe, and North America. China has the most research studies on paddy field utilization, followed by Japan, the United States, and India [26]. While case study areas in China mostly focus on the plains [27–29], and tend to be located within the northeastern plains [30], there is a lack of studies on IBPFU in the karst mountains of southwest China. Research methods commonly include standard deviational ellipse analysis, kernel density estimation, landscape index models, spatial overlay, autocorrelation analysis [6,31], etc. Morphological spatial pattern analysis (MSPA) methods have been applied relatively little in the evolution of cultivated landscapes. Existing studies on the spatiotemporal evolution of paddy fields have focused on plain areas using large spatial scales and short time series.

In the karst mountains of southwest China are scattered thousands of IBs comprising various flat intermountain landforms, including tectonic basins, river terraces, alluvial fans, and piedmonts. It is one of the world's three major contiguous karst distribution areas [32]. As another typical landform type relative to SL, IBs are the essence of karst mountain land resources in Guizhou Province, the center of food production and human activities in the mountainous karst areas [33]; they play an important supporting role in regional political, economic, cultural, and ecological construction [34]. In the mountainous region, flat land areas for cultivation are limited; thus, paddy field use is concentrated in IBs (locally named "bazi") formed by various geomorphological processes, which is important for food security in mountainous areas. Therefore, it is important to reveal the mountain–basin system's land use transition for sustainable land use in mountainous areas [25].

The main foreign studies on land use change in mountainous areas include rice farming efficiency in karst mountains [35], soil erosion response to land use change [36], effects of changes in paddy field use on soil bacterial communities in hilly and mountainous areas [37], the study of hydrological terraces in mountainous areas [38], the use of IBs and its effect on benthic fauna [39], the spatial assessment of forest cover and land use change [40], and impacts on landscape fragmentation on ecosystem services in mountain environments [41], etc. There is a relative paucity of research on land use in IBs. Chinese studies that focused on land use in IBs are related to factors influencing urban landscape patterns in IBs in southwest China [42], the spatial and temporal evolution of land use and landscape patterns in mountain–basin systems [33], and the spatial evolution of land use intensity and landscape pattern response in typical basins in Guizhou [43]. Previous studies lacked focused research on the spatial and temporal evolution of the core land use type of IBs, i.e., paddy field use over a long time series, neglected the spatial and temporal evolution and transformation of paddy field use in IBs, and could not answer the characteristics of land use transformation in IBs, so much so that they could not fully reveal land use evolution and transformation patterns in karst mountain areas in China and even globally.

Therefore, to fill these research gaps, this study took another geomorphic unit in the karst mountain region of southwest China relative to the SL, the Huishui Basin in Guizhou, as the research object. Huishui Basin is the largest intermountain basin in Guizhou Province in terms of distribution contiguity, and belongs to the karst basin landform; it exhibits typical characteristics of suburban areas. The land use type in the basin is mainly paddy fields, a typical intermountain basin for modern agricultural development in Guizhou Province. An in-depth exploration of the evolution and transformation characteristic patterns of its paddy field use can provide a basis for comprehensively revealing the transformation patterns of land use and the evolution of the human–land relationship

in the karst mountain areas of southwest China. Based on this, the objectives of this research were as follows: (1) to construct a framework for the study of the evolution and transformation of paddy fields in karst mountain basins; (2) to analyze the dynamic evolution process and transformation characteristics of the long time series of typical IBPFU as an example; and (3) to reveal the differences in land use transitions between IBs and SLs.

## 2. Materials and Methods

### 2.1. Theoretical Analysis Framework

In contrast to SLs, thousands of IBs exist in the karst mountains of southwest China. These basins are the main carriers of production and living activities in the karst mountains of southwest China [43]. The land use of IB mainly comprises traditional agricultural cultivation, and the mainland type is paddy fields [44]. With the rapid progress of social and economic transformation and urbanization via measures such as land consolidation, land transfer, planting structure, and industrial structure adjustment, the land use of IBs has evolved in a diversified manner, and land use function has gradually shown diversified transformation [45,46]. IB agriculture gradually developed from traditional to modern-scale agriculture, and IB land use exhibits gradual transformations. In contrast, the transformation of SL shows an evolutionary pattern of shrinking arable land and the restorative growth of forest land [47,48]. This study argues that while there are commonalities in the transition of traditional land use in the southwest mountainous IBs compared to the transition of SL use, there are also clear differences.

In the context of land use transitions and induced production substitutions in the IB, the use of paddy fields within the basin changed significantly in terms of area quantity and spatial patterns, especially, when the changes were very rapid after 2006 [49]. The conversion of paddy fields to other land types is evident, with the expansion of construction land within the basin, increased non-foodification, and increased fragmentation of arable land use such as paddy fields. The above phenomena indicate a fundamental transformation of predominant paddy field use and traditional agricultural production functions in the IB of karst mountains. Significant differences exist between the evolution of paddy field use, and the transformation of SL use. This study further argues that, in the context of multiple socioeconomic changes, the differences reveal the differential patterns of the transformation of IB and SL use in mountainous karst areas by studying the characteristics of the spatial and temporal evolution of IBPFU in karst mountain areas, such as stage, diversity, reversibility, and irreversibility. Accordingly, based on the theory of land use transformation [50], this research constructed a research framework on the evolution and transformation of IBPFU in karst mountain areas from the perspective of structural and functional evolution (Figure 1), and explored methods that empirically prove the land use transformation of typical landform units via the evolution of core land use types. Firstly, we summarized the characteristics of SL and IB land use change by carrying out a literature review; secondly, we analyzed the spatial layout, number of areas, paddy use type, characteristics of paddy land use transition, and the structural to functional evolutionary transition of IBPFU using the net change rate of paddy fields, dynamic degree of land use, moving window method, MSPA model, and the land use transfer matrix method. Based on this, we summarized the spatial diversity, type diversification, temporal stages, and reversible and irreversible characteristics of the evolution process of IBPFU in mountainous karst areas. This study also analyzed the driving mechanisms of the evolution of paddy fields from various aspects, such as natural resource endowment, socioeconomic conditions, regional development policies, and urbanization development in IBs. Under the trade-off of social benefits, economic benefits, and ecological construction, single-function agriculture in the IB can be developed into multi-function agriculture, and traditional agriculture can be transformed into modern agriculture. Ultimately, the goals of regional food security, the optimal use of paddy fields, and sustainable development in mountainous karst areas can be achieved.

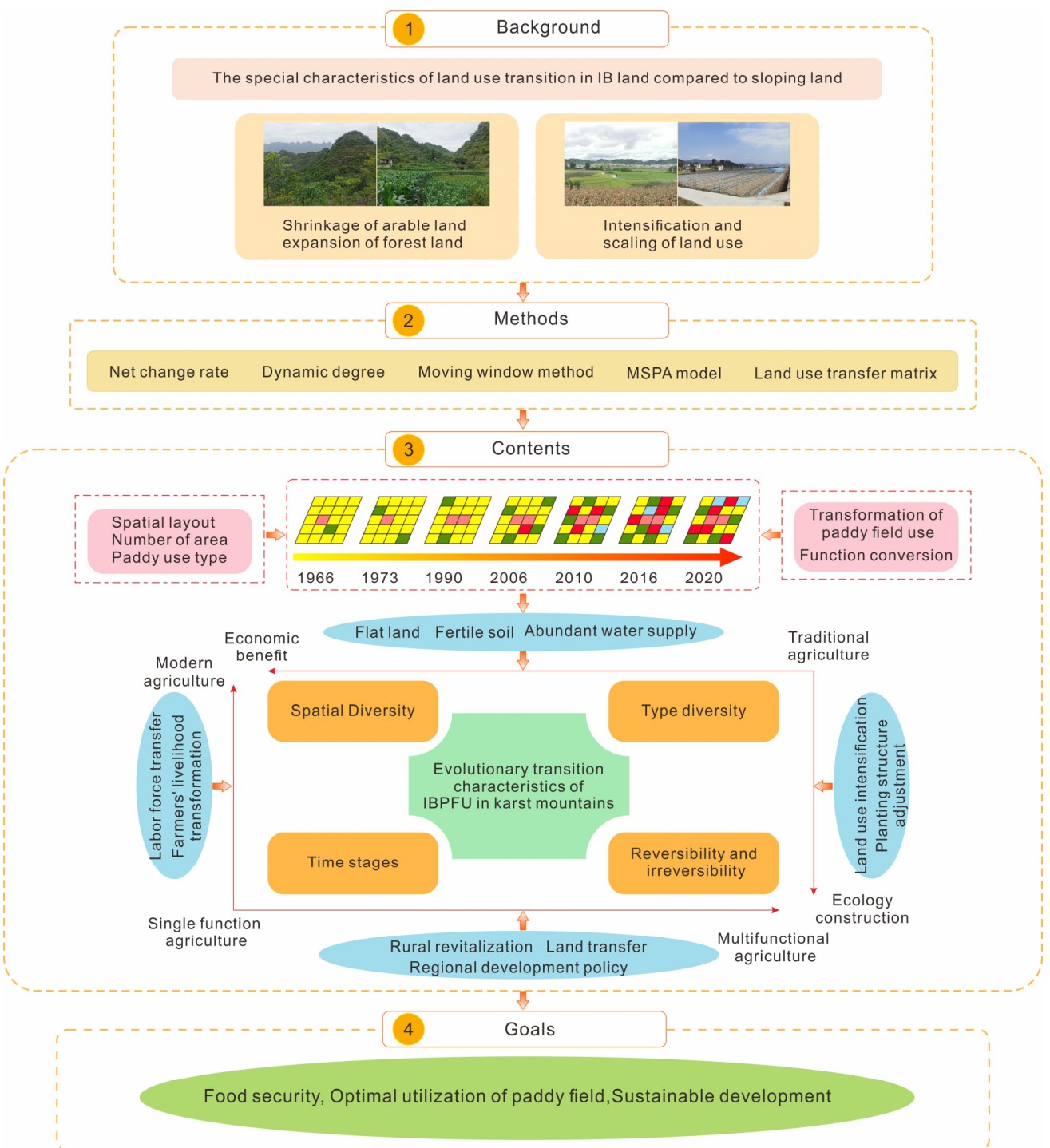

**Figure 1.** Analysis framework of the transformation in the IBPFU in the karst mountains of southwest China.

### 2.2. Study Area

The Huishui Basin is located in the northwestern part of Huishui County, Guizhou Province, southwest China, adjacent to Huaxi District of Guiyang City in the north, with well-developed transportation and significant location advantages. It is the center of economic development in Huishui County. This IB is the only basin with an area of more than 100,000 mu (mu, the Chinese unit of land measurement that is commonly 666.7 square meters) in Guizhou Province, with a total area of 85.71 km$^2$; it is a typical karst basin landform, with a minimum elevation of 871 m and a maximum elevation of 1062 m. The

average annual temperature is 14–16 °C, its yearly precipitation is 1178.2 mm, and the frost-free period is about 280 days. According to the international system of soil classification, the soils in the study area are mainly sandy, sandy loams, and loamy soils. The IB is formed by the alluvial accumulation of the Lian River, running from north to south. It is also known as the Lian River Basin, a typical and representative basin in the karst mountains of southwest China (Figure 2). With abundant water resources and fertile soil, the IB has long been planted mainly to grow rice and other traditional crops, and is one of the vital rice production bases in Guizhou Province. However, in recent years, due to the low economic returns from paddy field cultivation of food crops, most of the traditional rice cultivation in the IB has gradually shifted to modern-scale facility agriculture, such as greenhouse vegetables, rice-crayfish culture, fruit base, and the flower and seedling industry, and economic forests; these have improved income, and led to significant changes in land use in the IB. The IB has gradually shifted from traditional agriculture based on paddy fields to modern-scale facility agriculture [51], becoming a vital demonstration base for modern high-efficiency agriculture in Guizhou Province.

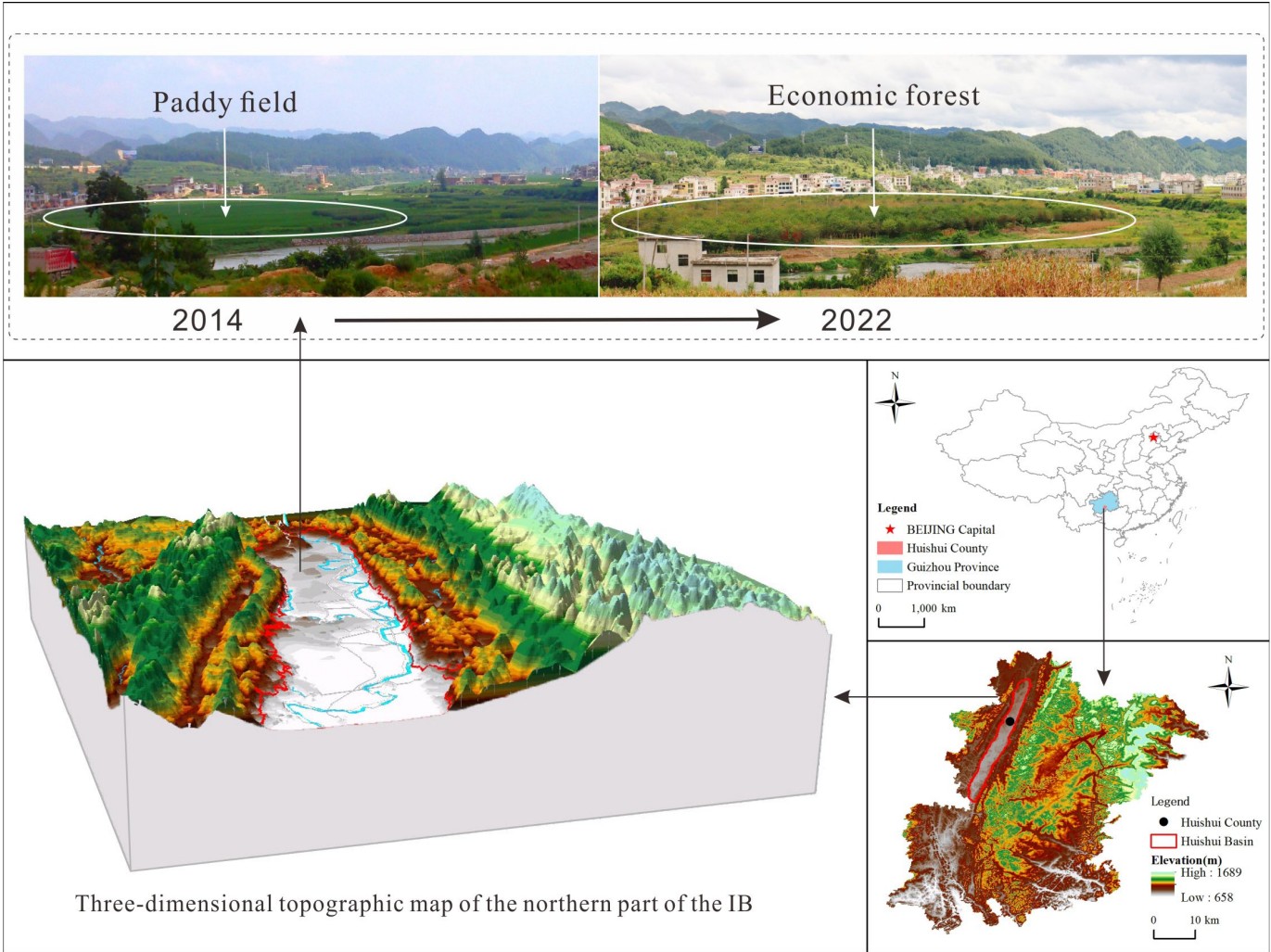

**Figure 2.** Location and topography of the study area (The pictures in this figure were taken by the author in the Huishui basin in Guizhou Province in 2014 and 2022).

### 2.3. Data Source and Processing

Land use data of the study area were obtained from high-resolution remote sensing images, such as panchromatic American KeyHole satellite images, SPOT images, ALOS images, and topographic maps, including seven periods of land use data from 1966 to 2020.

The data resolution, acquisition time and sources of each period are detailed in Figure 3. Based on the ArcGIS 10.2 software platform, the accuracy correction, coordinate system conversion, and manual visual interpretation of each period image or topographic map were processed. The land use data were modified and verified with several field survey data, and the accuracy rate reached over 95%. The socioeconomic industry data involved in the text is from http://www.gzhs.gov.cn/ accessed on 18 December 2022. The classification of land use functions was established according to the "Land Use Status Classification" (GB/T21010-2017), promulgated in 2017, and based on the theoretical classification method of the "production-living-ecological" function [52] combined with the characteristics of land use in karst mountain basins and research needs, as shown in Table 1.

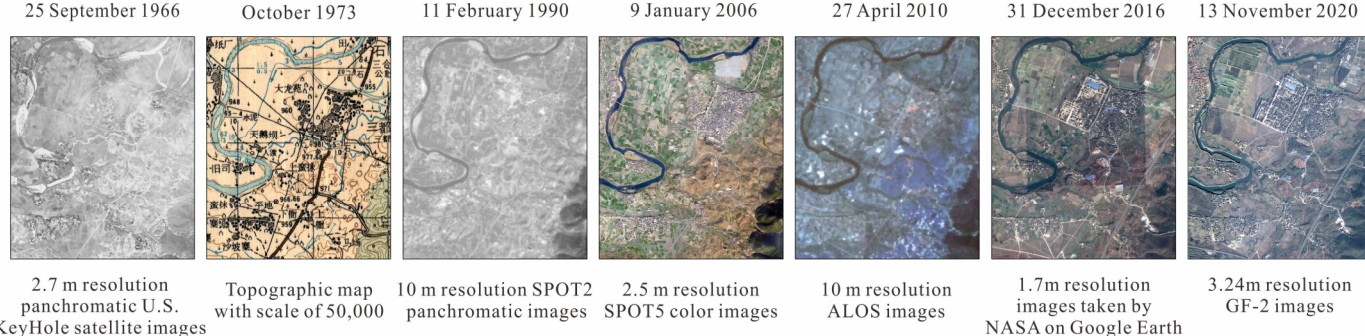

**Figure 3.** Source and resolution of data in the study area.

**Table 1.** Land use function of the classification system of IB.

| Primary Classification | Secondary Classification | Indicator Layer (Land Use Pattern) |
|---|---|---|
| Life functions | Residential and living function | Residential and living land (mainly including residential land, public infrastructure land, land under development and construction, road land, and tourism land) |
| Production function | Traditional agricultural production functions | Paddy fields, dry land (mainly including land for rice cultivation, land for corn cultivation, land for wheat cultivation, and land for potato cultivation) |
| | Modern-scale facility agricultural production function | Modern-scale agricultural land (mainly including land for greenhouse vegetables, open-air vegetable base contracted by the company, lotus root planting base, Zizania latifolia planting base, strawberry planting base, watermelon planting base, grape planting base, flower base, peach and plum planting land, and the large area of land covered with mulch) |
| | Industrial production function | Industrial and mining storage land (mainly including industrial land, mining land, and storage land) |
| Ecological function | Natural ecological function | Ecological land (mainly including forest land, shrub land, grassland, river water surface, reservoir water surface, pond water surface, ditch, idle land, artificial turf planting base, and landscape seedling planting base) |

*2.4. Calculation of Indicators*

2.4.1. Land Use Change Measurements

(1) The net change rate of paddy fields was used to reflect the extent of inflow or outflow of paddy fields in a certain period, generally expressed as the net change ratio of paddy fields to the total amount of paddy fields at the beginning of the study period [6].

$$N = \frac{S_b - S_a}{S_a} \times 100\% \tag{1}$$

N is the net rate of change of paddy fields. $S_a$ and $S_b$ are the areas of paddy fields in the study area at the beginning and end of the study, respectively.

(2) The dynamic degree of land use is the change in the quantity and the rate of change of the paddy fields during the study period, reflecting the stability of the paddy fields' use of resources. The higher the dynamic degree, the less stable the paddy fields' use of resources [53].

$$K = \frac{\Delta U_i}{t \cdot s} \times 100\% \tag{2}$$

K is the dynamic degree of land use types, $\Delta U_i$ is the amount of change in the area of paddy fields in the study area, $s$ is the area of the paddy field at the beginning of the period, and $t$ is the time interval year.

(3) The land use transfer matrix describes the initial and final transfer between land use types and the transfer rate between land use types in a certain region in a certain period [54].

$$S_{ij} = \begin{bmatrix} S_{11} & \cdots & S_{1n} \\ \vdots & \ddots & \vdots \\ S_{m1} & \cdots & S_{mn} \end{bmatrix} \tag{3}$$

$S$ is the area, $i$ and $j$ are the land use types before and after the transfer, respectively, $n = 1, 2, 3 \ldots \ldots$, $m = 1, 2, 3 \ldots \ldots$, and $S_{ij}$ is the area after the conversion of type $i$ to type $j$ before the change.

### 2.4.2. Moving Window Method

The moving window method involves selecting a window with an appropriate side length, and moving from the upper left corner of the study area. Each time it moves one raster, the value of the landscape indicator within the window is calculated and assigned to the center raster of the window. Finally, the raster map of the landscape indicator is output [55]. In this research, the moving window method with Fragstat 4.2 software was used to determine the spatial distribution of the percentage of basin paddy fields area by screening a rectangular moving window with a side length of 250 m, and selecting the percentage of landscape occupied by patches (PLAND) indicator.

### 2.4.3. Morphological Spatial Pattern Analysis

The landscape index model is primarily used in the analysis of patch fragmentation. The MSPA model is a spatial pattern analysis method proposed by Vogt et al. [56]. It allows the identification, segmentation, and analysis of raster images to explain multiple landscape morphological changes in the study area, resulting in an accurate landscape structure at the image element level [56]. According to the land use data of the study area, paddy fields were extracted as the foreground data for MSPA analysis, and other non-paddy fields data were used as the background data and assigned as 2 and 1, respectively. Through the analysis of Guidos Toolbox 2.8 software, seven landscape types of paddy fields could be obtained, and their meanings are shown in Table 2.

**Table 2.** Features and definitions of landscape types based on the MSPA.

| | Landscape Type | Characteristics and Meaning |
|---|---|---|
| | Core | The larger habitat patches in the foreground image represent large paddy patches, and have the greatest influence on the overall spatial pattern of paddy use. |
| | Islet | It is isolated, with broken patches that are not connected to each other. |
| | Perforation | The internal object perimeter, the transition area between the core and non-paddy field patches, and is the edge of the inner patches. |
| | Edge | It is the transition between the core and main non-paddy landscape areas. |
| | Loop | Connecting the narrow area of the core area helps to connect within the core paddy field. |
| | Bridge | It is a collection of image elements connecting different cores. |
| | Branch | It connects the core area paddy fields and other paddy field types. |

Legend: Bridge, Branch, Core, Islet, Perforation, Edge, Loop, Background

## 3. Results

### 3.1. Structural Evolution Characteristics of Basin Paddy Use

3.1.1. Quantitative Evolutionary Characteristics of Basin Paddy Use

From 1966 to 2020, the quantity of IBPFU varied significantly, exhibiting a general decreasing and increasing trend, and resource stability gradually weakened. Up until 2016, the area of IBPFU decreased from 52.01 km$^2$ to 21.23 km$^2$, with a total decrease of 30.78 km$^2$; this decrease accounted for 59.18% of the original paddy field area; among the declines, the largest was in 2006–2010, which exhibited a decrease of 9.23 km$^2$; the paddy field area turned to an increasing trend in 2016–2020, with an increase of 7.34 km$^2$. The net change rate and the dynamic degree of the IBPFU area were generally high, especially since the change after 2006 was particularly significant. It can be observed that the stability of paddy field resources in the study area was poor, especially since the stability of paddy field resources after 2006 was the weakest (Figure 4).

3.1.2. Spatial Evolution Characteristics of IBPFU

The spatial distribution of the percentage value of paddy field area from 1966 to 2020 gradually changed from contiguous high values to scattered low values (Figure 5). Before 2006, the spatial distribution of the percentage value of the paddy field area did not change significantly and maintained a high percentage of values above 90%. Starting from 2006, the high values of the percentage area of paddy fields began to contract, and the distribution became gradually dispersed. The contraction of the town area and the northeast area in the central part of the study area was more significant, indicating that the reduction in the paddy field area in the study area occurred mainly around the town, and it was difficult to revive this part of the paddy field. In the southwest area of the study area, the distribution of the percentage area values of paddy fields contracted and dispersed, and then showed a clear trend of expansion, but the percentage area values of their expansions were low, indicating a discontinuous spatial distribution of paddy fields.

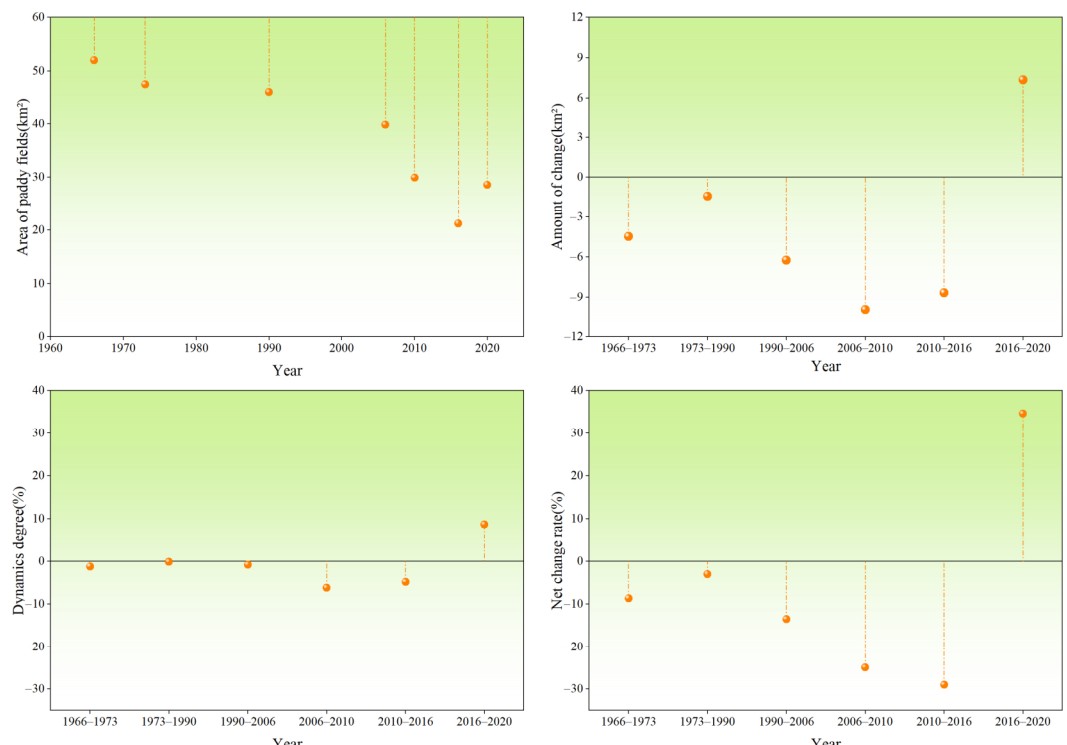

**Figure 4.** IBPFU area and its change from 1966 to 2020.

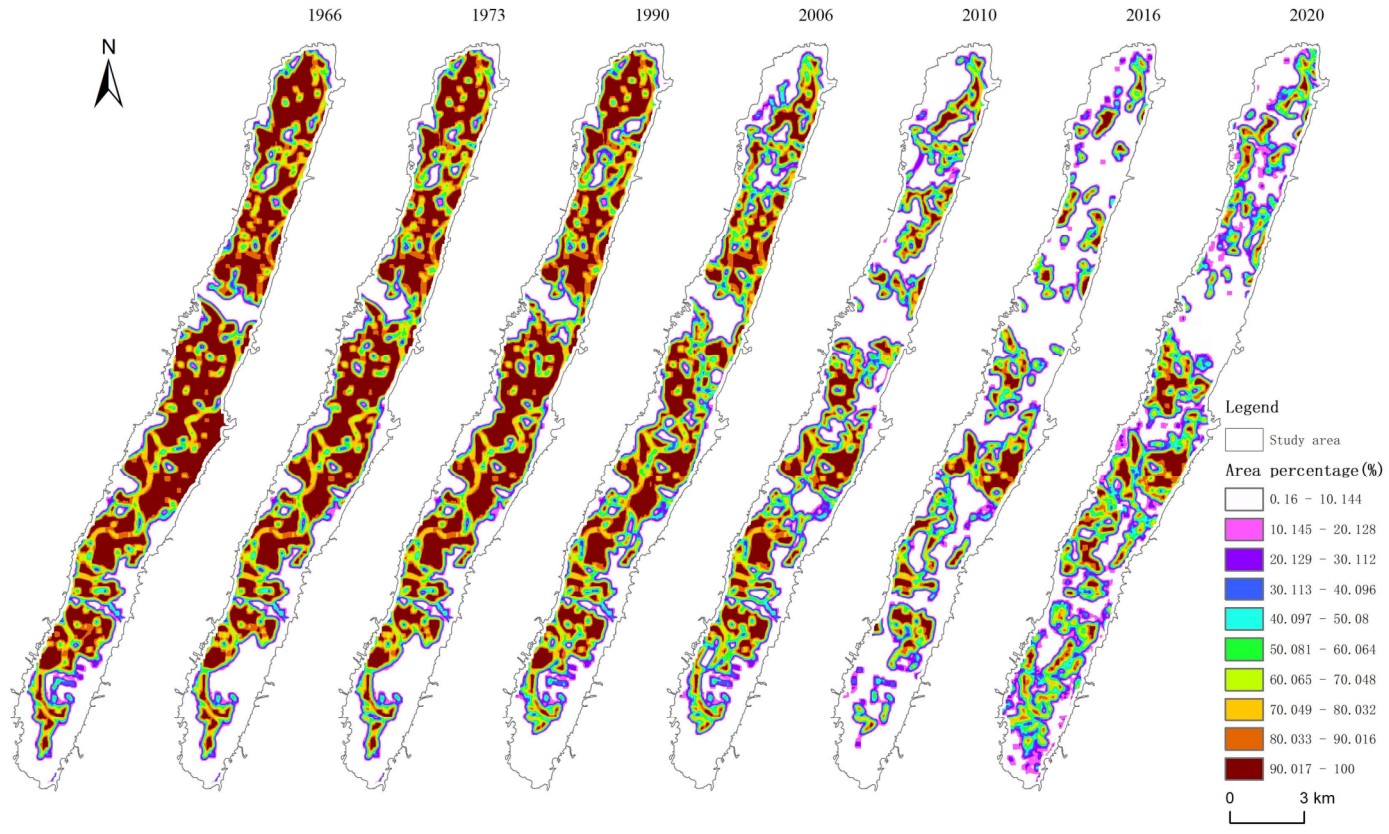

**Figure 5.** Spatial distribution of percentage area of paddy fields, 1966–2020.

### 3.1.3. Spatial Morphology Pattern Evolution of IBPFU Based on MSPA

Overall, the trends of the seven paddy landscape types of the IB were more significant in the last 10 years than those in the previous 44 years, and the spatial landscape patterns of IB paddies have become more complex and fragmented. Specifically, the core paddy field area of the IB decreased from 35.06 km$^2$ in 1966 to 6.13 km$^2$ in 2020, and the spatial distribution changed from agglomeration to dispersion. With the disappearance of patches in the core paddy field area, the perforation and edge areas also decreased significantly, while the area with the islet showed an increasing trend, and the area with bridges, branches, and loops showed an increasing, then decreasing, and then increasing trend. Among them, the area with bridge paddy fields re-increased by a large amount, and is mainly distributed in the southwest of the IB, showing a trend of paddy field recovery in the southwest direction away from the town (Figure 6).

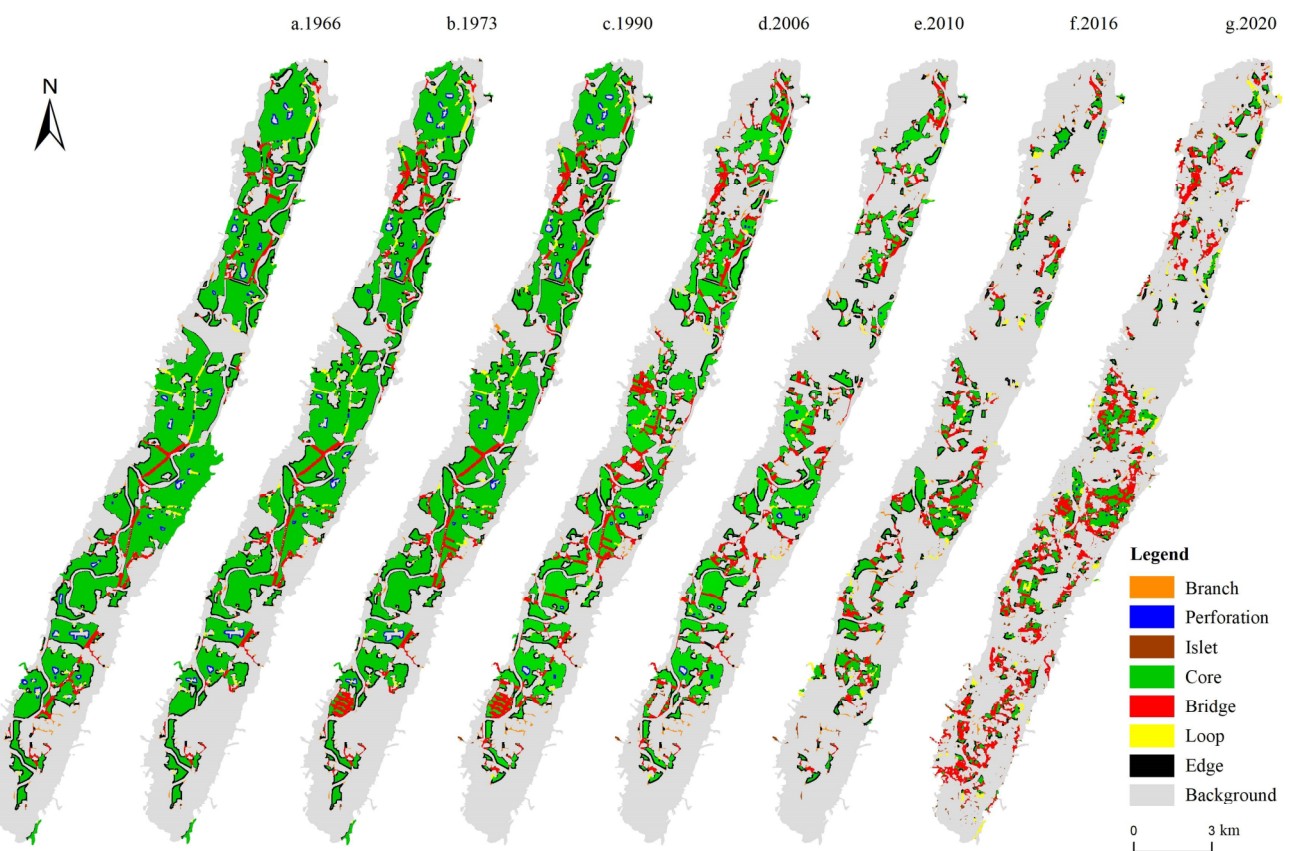

**Figure 6.** Morphological spatial patterns of the IBPFU during 1966–2020.

### 3.2. Functional Transformation Characteristics of IBPFU

The transfer characteristics of the paddy fields in the IB varied during the study period. In terms of the type of transfer (Figure 7), the transfer from paddy fields to dry land and residential and living land was predominant in 1966–1973 and 1973–1990. At this time, the land use of the IB was single. The land use functions mainly comprised traditional agricultural production and residential and living functions, in order to meet the basic survival needs of farmers. From 1990 to 2006, the transfer of paddy fields in IB was dominated by a transformation from paddy fields to modern large-scale agricultural lands, followed by a transformation to residential and living lands and ecological lands.

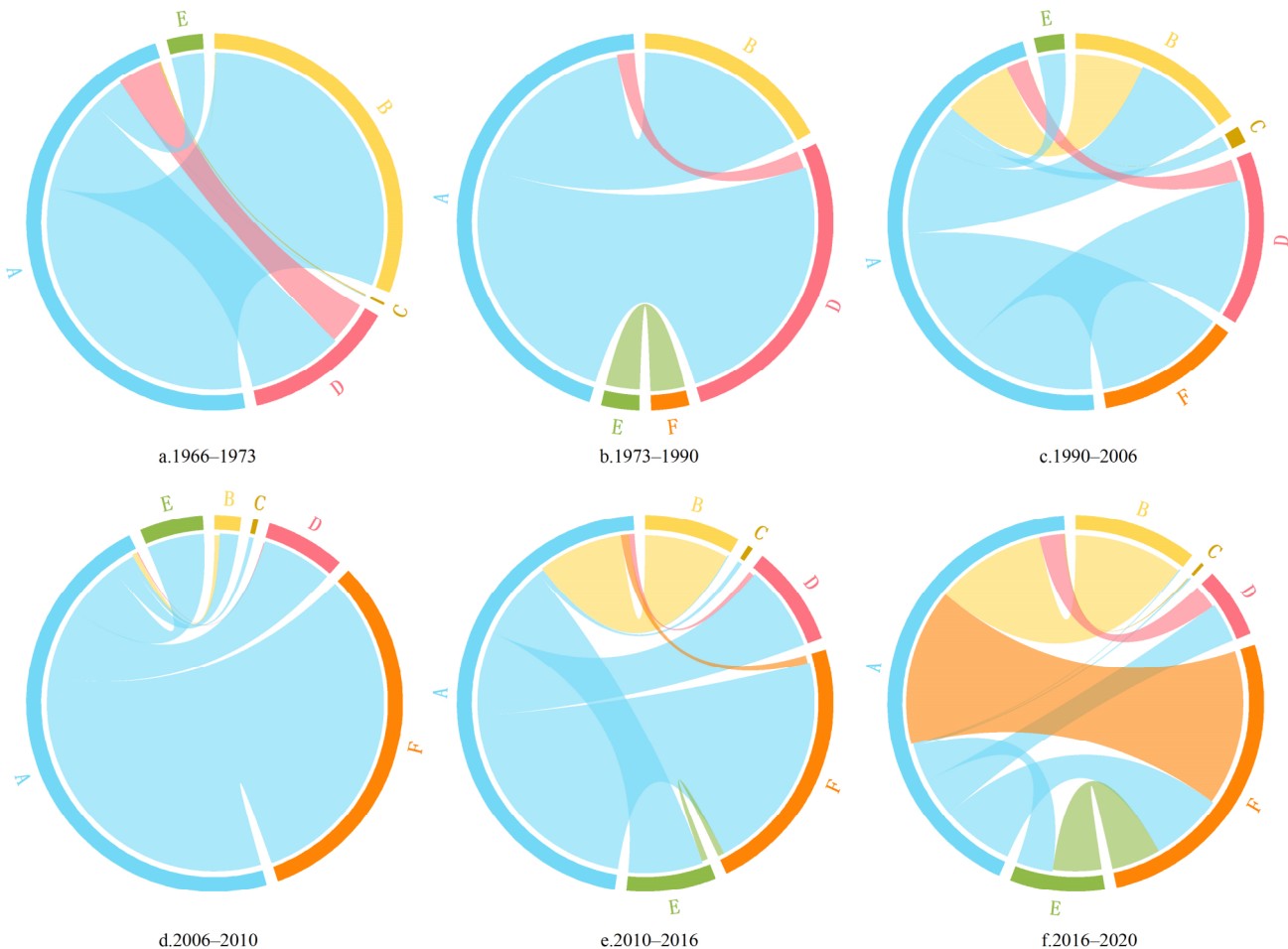

**Figure 7.** Transferred area of IBPFU and its percentage from 1966 to 2020. A, B, C, D, E, and F in this figure represent the paddy field, industrial and mining storage land, residential and living land, ecological land, and modern-scale agricultural land, respectively.

With social and economic developments and the implementation of policies such as the new urbanization plan and the rural revitalization strategy, the land use transformation of the IB in Guizhou Province has accelerated, and the land use function has gradually diversified with respect to a change in people's demands, while simultaneously prompting a sharp decline in paddy fields in the study area. Up until 2016–2020, some modern-scale agricultural lands reverted to paddy fields under the implementation of cultivated land protection and food security policies.

In terms of spatial distribution (Figure 8), the most significant changes were observed around Huishui County and north of the town in the first three periods, and the range of changes gradually extended to the entire study area in the last three periods; the paddy field transfer points in the central and northern parts of the study area gradually disappeared, indicating that the transfer of paddy fields to residential and industrial land is irreversible. Overall, the land use of IB in the karst mountains is mainly based on production and living functions. The scale of modern-scale agricultural production functions in IB is expanding, but the spatial distributions are not continuous.

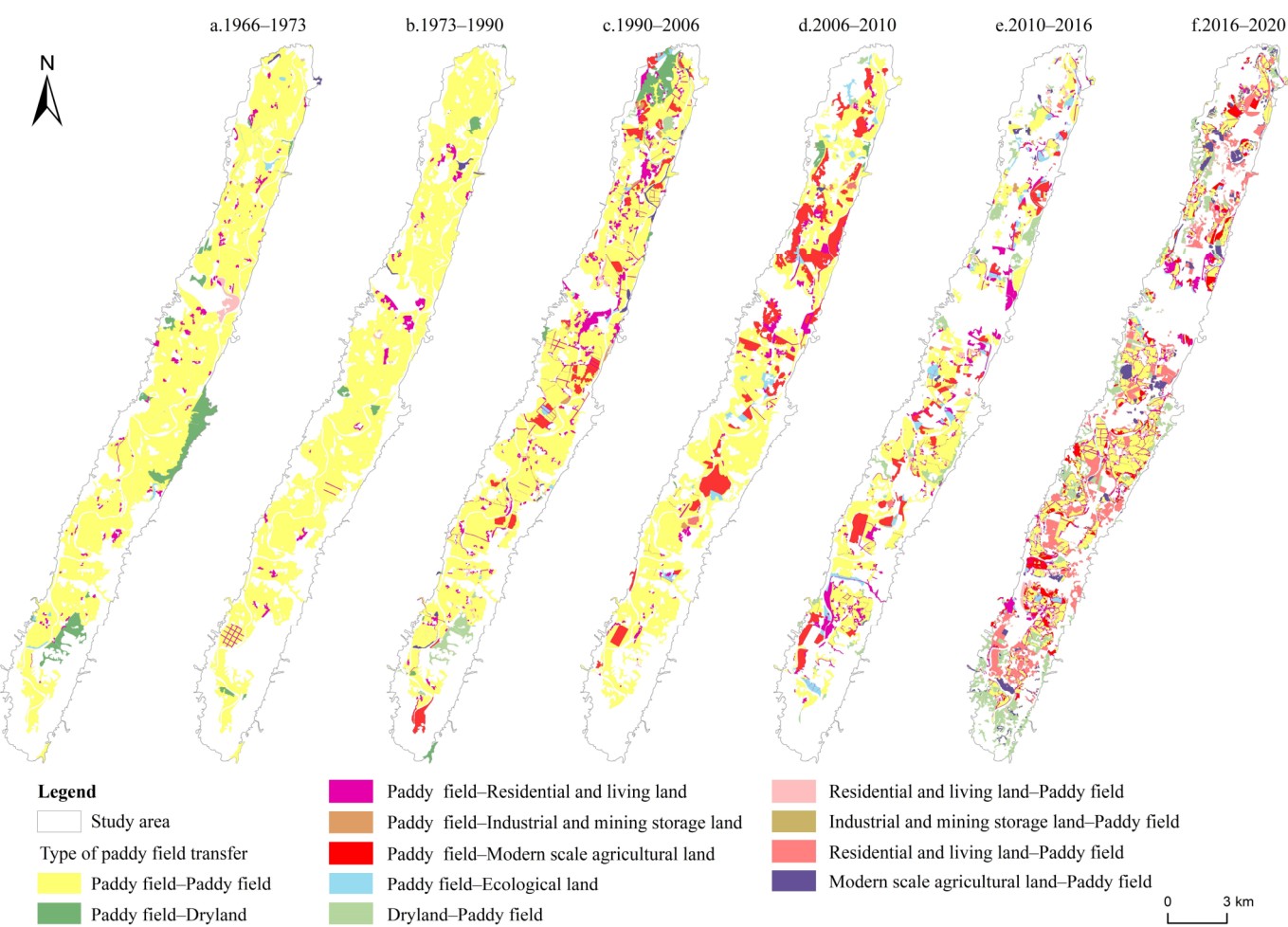

**Figure 8.** Spatial distribution of the transfer of IBPFU.

### 3.3. Evolutionary Stage Model of IBPFU

Studies have shown that the land use transformation of IBs in Guizhou Province corresponds to its socioeconomic development stage [49]. In this study, drawing on international research results on forest transition models and land use transitions in IBs [51,57,58], we established a model of the evolutionary stages of IBPFU in the study area based on the study of the characteristics of IBPFU transfers; the number of paddy fields in the study area in 1966, which was used as the base period paddy fields; and the proportion of the transfer of paddy fields to other land types in each period relative to the area of paddy fields in the base period (Figure 9). The model showed that at different stages of socioeconomic development, IB land use was different, and the area of IB paddies showed different change trends at different periods; the IB land use function changed from a single traditional function period to a multi-functional gradual transition period, multi-functional development enhancement period, and multi-functional intensive use period. With enhanced intensive land use in the IB, the implementation of cultivated land protection and food security policies has increased the area of IBPFU.

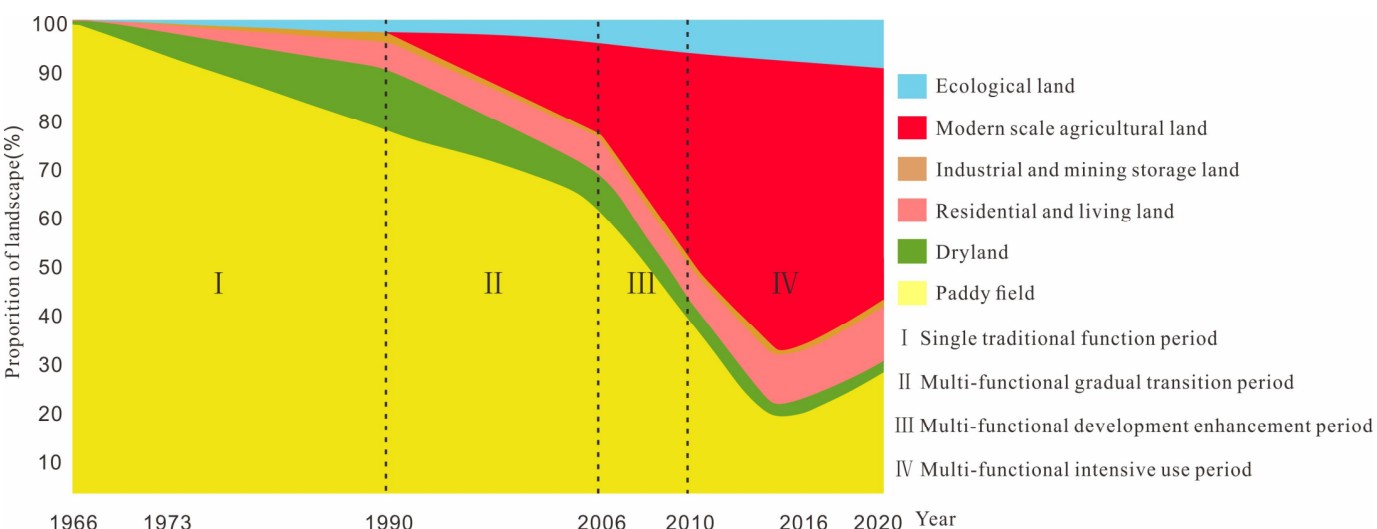

**Figure 9.** The evolutionary stage model of IBPFU.

### 3.4. Spatial Diversity in the Evolutionary Stages of IBPFU

Five typical areas, mainly located in the northern, central, and southern parts of the study area, were selected in combination with the characteristics of paddy field evolution types, in order to analyze in-depth the evolutionary process of paddy fields in different study areas in the last 54 years (Figure 10).

(1) Northern region A: The northern part of the IB study is 28 km away from the provincial capital city of Guizhou Province, and has significant geographical advantages. The main process of paddy field evolution comprised paddy field–dry land–industrial and mining storage land, with 2006 and 2016 being the main turning points in its evolution. Before 2006, the evolution mainly comprised a change in the structure of cultivated land use, and its land use function remained unchanged. After 2016, most changed paddy fields were difficult to restore and were irreversible, leading to a shift in land use from mainly traditional agricultural production to the main industrial production function.

(2) Central region B: The primary process of paddy field evolution was paddy field–residential and living land. From 1966 to 2020, the paddy fields gradually shifted to commercial service areas; residential land; areas with science, education, culture, and health sectors; transportation land, etc., at the center of Huishui town, and the distribution was concentrated and continuous. Restoring the original paddy fields after the change was difficult, and the trend only decreased but did not increase, exhibiting irreversibility. Its land use function has gradually changed from traditional agricultural production to residential and living functions.

(3) Southern regions C and D: The main paddy field evolution process comprised paddy field–modern-scale agricultural land–paddy fields, and paddy field–modern-scale agricultural land–ecological land. The 2006 and 2010 years were the main turning points in its evolution. Region C gradually shifted from paddy field use to modern-scale agricultural lands, such as nursery gardens, flowers, and fruits, since 2006, and the scale has gradually expanded; moreover, all of its changed paddy fields can be restored, exhibiting reversibility. The large-scale transfer of paddy fields in region D to modern large-scale agricultural land only began in 2010, and it partly reverted to paddy fields in 2020. The land use function of these two regions has shifted from traditional agricultural production to modern-scale facility agricultural production and natural ecological functions.

(4) Southern region E: The main processes of paddy field evolution comprised paddy field–dry land–paddy field and paddy field–modern-scale agricultural land–paddy field, and these were reversible processes comprising the interconversion of paddy fields with

modern-scale agricultural land and dry land. The area is far from Guiyang City, with few original paddy fields, and mainly comprised traditional agricultural cultivation.

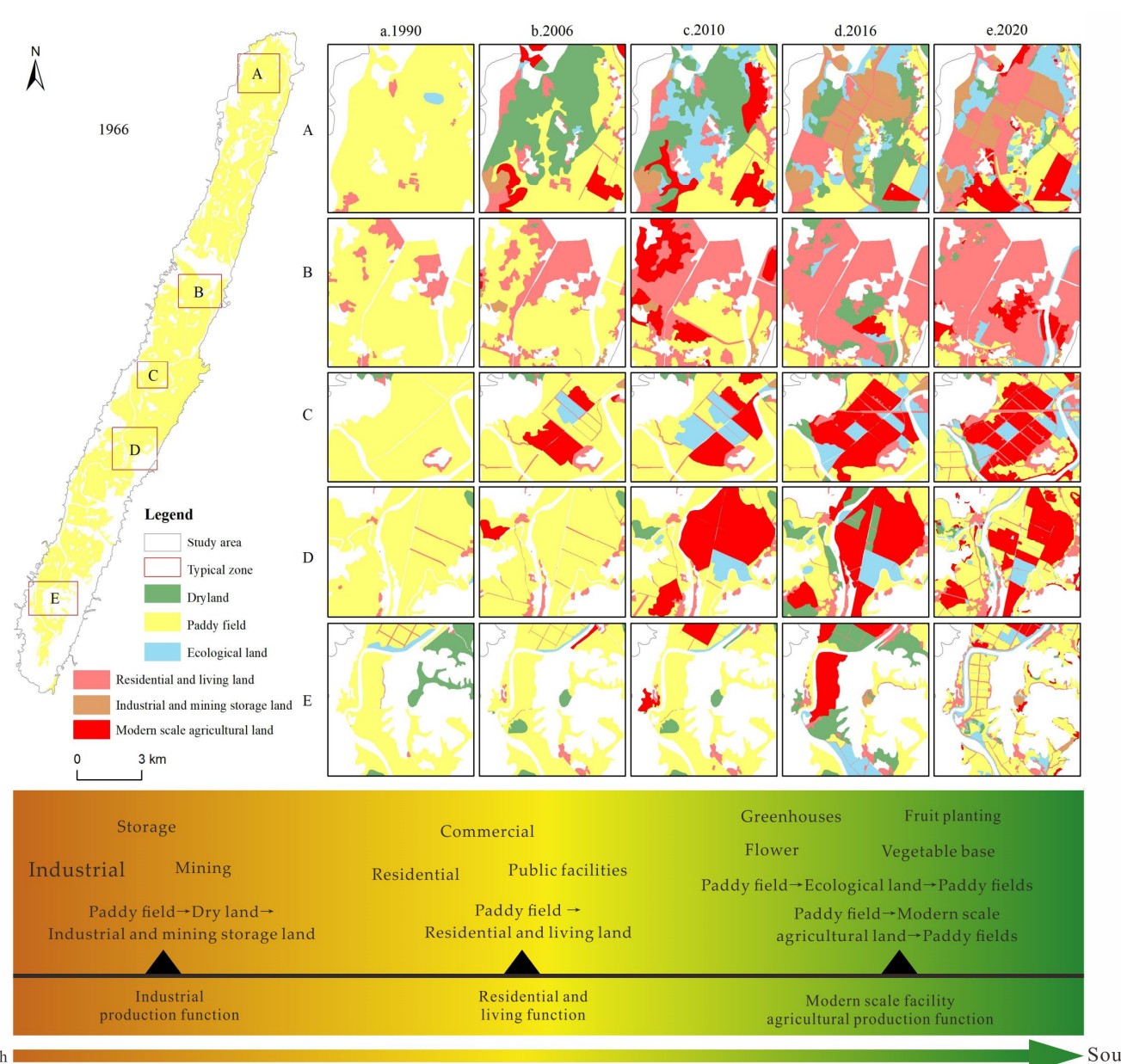

**Figure 10.** Spatial distribution of IBPFU and their typical evolution types.

In summary, the studied IBs have significant spatial differences with respect to the evolution of traditional agricultural production functions based on paddy field use in 1966–2020; spatial diversity in the types of paddy field change; reversibility and irreversibility in the change process; and regional inconsistency during the time of change. The closer the IB was to Guiyang from the southern to northern regions, the earlier the land use transition began, and the degree of change was more significant. From the above patterns and characteristics of the evolutionary process of paddy fields, it can be concluded that 2006, 2010, 2016, and 2020 were the main time points of the change in IBPFU.

## 4. Discussion

### 4.1. Evolving Patterns of IBPFU

Studying the evolution of IBPFU, especially the transformation of the utilization function in recent years, reflects the land use changes in the process of transformational development and the rural revitalization of Chinese villages. The evolutionary pattern of IBPFU has a temporal phase, and this is embodied in the fact that the area of IBPFU shows different trends during different periods. The spatial diversity of the evolutionary patterns of IBPFU is reflected in the different evolutionary types and evolutionary processes of IBPFU change in different locations. The evolutionary pattern of IBPFU is diversified and characterized by the evolution of traditional agriculture in the IB, mainly paddy fields, to modern agriculture, service industries, and the breeding industry. The evolution of traditional to modern agriculture is reflected primarily in the rotation of rice and rape, morel mushrooms, and other cash crops, as well as the rice and shrimp crop model; most farming industry raises aquatic animals such as fish, shrimp, and duck. Paddy fields used in modern agriculture are mainly planted with cash crops such as Zizania latifolia and lotus roots, while the use of paddy fields that are mainly planted with cash crops such as fruit forests, plants and flowers, and greenhouse vegetables has transformed. What also led to the transformation of paddy fields was the development of service industries, which indicated that the IBPFU has transformed to implement economically efficient production methods (Figure 11). Although the area of the IBPFU was restored later, some paddy fields are planted with cash crops instead of food crops, such as Zizania latifolia and lotus roots, which will inevitably have a long-term impact on regional food security. Therefore, during the evolution of IBPFU, attention should be focused on protecting cultivated land and ensuring food security; moreover, the ecological and environmental effects caused by the evolution of paddy fields should not be ignored [7,26].

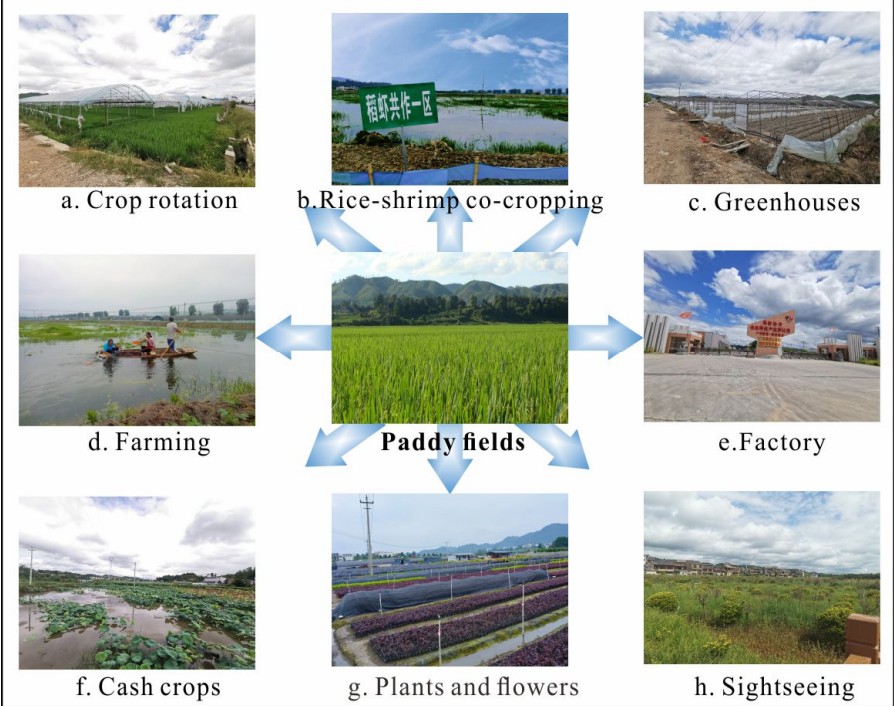

**Figure 11.** Evolutionary pattern of paddy field use. The pictures in this figure were taken by the author in the Huishui basin in Guizhou Province in 2022.

From the evolution pattern of paddy land use, there are commonalities and differences between the evolutionary pattern of paddy field use in this research and other regions in China and abroad. In an earlier study, Barai et al. [59] analyzed land use patterns in

shrimp and rice-producing areas in southwestern coastal Bangladesh; it was observed that the expansion of shrimp cultivation in the region mainly occupied paddy areas. In northeastern Thailand and the Laos–China Border region [60], the paddy fields primarily shifted toward modern agricultural infrastructure, such as rubber plantations. In China's more economically developed regions, the paddy fields are mainly occupied by urban construction, which is difficult to restore, such as the Three Gorges Reservoir Region [61], the Hangjiahu Plain [62], and the Yangtze River Delta [63]. However, the evolutionary patterns of paddy fields in these areas are relatively one-dimensional, while the evolutionary patterns of IBPFU in this research are more diversified.

### 4.2. Transformation Drive Mechanism of IBPFU

The change in IBPFU results from the interaction between human and natural factors [64], and the natural conditions within the basin are relatively stable; thus, the influence of socioeconomic and human factors on the change in IBPFU was mainly analyzed (Figure 12). Changes in the IBPFU of the karst mountains of southwest China have different dominant factors at different periods. Before 2016, the change in IBPFU was mainly influenced by regional urbanization and market demands, which led to a decrease in the area of paddy fields within the basin. After 2016, the reduction in the area of IBPFU was effectively curbed via the government's policy guidance.

In particular, with social and economic development, the implementation of policies such as the new urbanization plan, and the rural revitalization strategy, a large amount of cultivated land has been occupied by constructing towns and infrastructure. At the same time, driven by market demand, many residential lands within the basin have been internalized for farmhouse enjoyment, and farmhouse hotels and their residential and living functions have gradually shifted to production–living functions [45,49]. The traditional agricultural production functions of IB have been transformed into modern-scale facility agricultural production and tourism functions. Rural tourism and modern agriculture have become leading industries in the IB. The main body of IB land use has changed from main farmers to mainly local residents, companies, and outsiders, leading to the intensification of IB land use, diversification of land use functions, and substantial changes in the spatial landscape of IB, resulting in a drastic decrease in the area of IBPFU. The relationships between IBPFU and towns, paddy fields, and cash crops are negatively correlated, with the negative correlation between paddy fields and towns increasing, then decreasing, and increasing again; moreover, the negative correlation between paddy fields and cash crops shows a gradual increase. After 2016, paddy field areas began to gradually increase due to the policies driven by the "Implementation Plan of Huishui County Relocation Project," the "Measures for Protecting Land Planted in Basin Areas Over 500 Mu in Guizhou Province," and the "One Basin, One Policy" program. In previous studies, few have analyzed the specific causes of changes in paddy use from different stages. The government of Kerala, India—studied by previous authors—curbed the reduction in regional paddy areas by enacting the "Kerala Conservation of Paddy Land and Wetland Act" [4], which is similar to the conservation of paddy areas using the government's policy guidance, outlined in the latter part of this study. However, in Japan [65] and some European countries [66], population and climate change are the main causes of paddy field change, and low agricultural benefits directly lead to paddy field abandonment, indicating differences in the factors influencing the change of paddy fields within different regions.

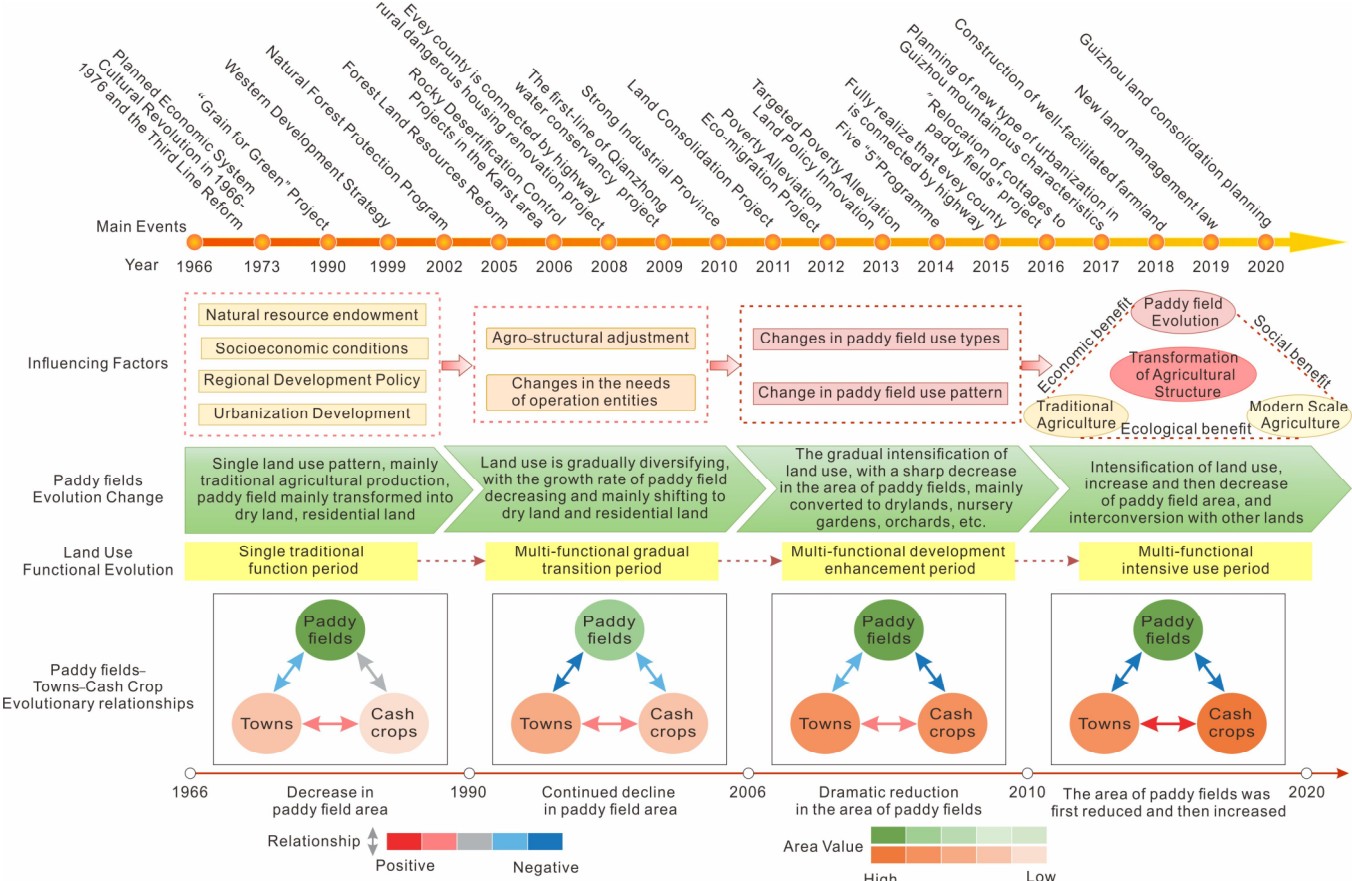

**Figure 12.** The driving mechanism of transformation of IBPFU (part of the structure and color modified by Wu [67]).

## 4.3. Comparison of the Transition of IBPFU and the Transition of Cultivated SL Use

The evolutionary pattern of the transition of typical IBPFU studied in this research is a snapshot of the evolution of IBPFU in the karst mountains of southwest China, not only showing that the characteristics of the transition of IBPFU have commonalities with the transition of SL cultivated under different socioeconomic development contexts, but that they also have specificities. The commonality is reflected in their land use patterns, which transform (by different means) into more economically efficient production methods. In the transition process, there are obvious temporal stages, spatial diversity, and diversification of evolutionary patterns, which are, in essence, a transformation in the diversity of land use functions [68]. The specificity reflects the close relationship between SL and its terrain slope in the transformation process, showing an increase in the reduction in cultivated SL area relative to an increase in the slope gradient [69]. Different topographic positions are planted in different ways. Higher topographic positions are only planted with fruit trees, while lower slopes are planted with intercropping fruit trees such as apples, pears, and cherries, with food crops such as corn. The transformation of IBPFU is mainly related to its distance from towns and major roads. When the IBPFU is closer to a town, the paddy fields are mainly transformed into construction land and modern agricultural land, and their land-use-saving ability and intensification intensity continually increase; at the same time, as IBPFU bears the heavy burden of food security in mountainous areas, the transformation of IBPFU requires a trade-off between regional socioeconomic development, ecological protection, and food security (Figure 13). Therefore, in China, developing a policy for protecting paddy fields according to local conditions is crucial in order to rationally plan cultivated land use. In particular, in the mountainous areas of southwest China, special

attention should be focused on protecting the karst mountain IB of cultivated land, thus guaranteeing food security and achieving sustainable development.

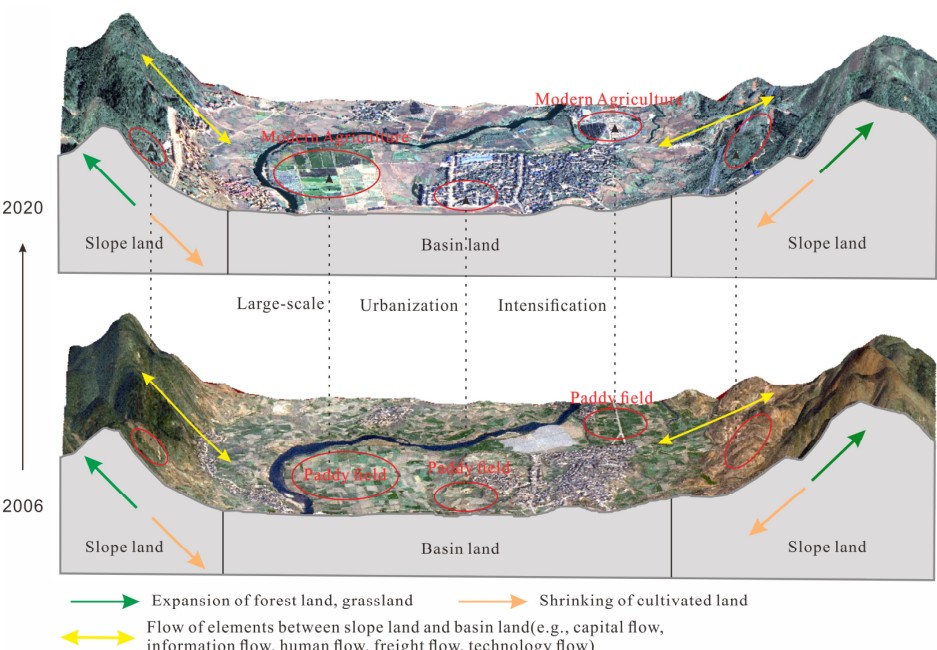

**Figure 13.** Transformation pattern of slope cultivated land and IBPFU in karst mountains of China.

**5. Conclusions**

Via the study of the core land use types of IB, this study showed that the land use function of IB in the mountainous karst areas of southwest China has undergone an obvious transformation, and there are commonalities and differences between the land use transformation of IB and mountainous SLs. This research studied the evolutionary transition characteristics of the IBPFU case, and can provide a better reference for the optimal management of sustainable land use in mountainous areas in China and globally; moreover, the study confirmed that the regional land use transition could be revealed by the evolution of its core land use types, and the core land use type comprises a typical landform unit over a long time series. The framework and methods of this research could equally apply to other regions. There are some shortcomings in this study: firstly, the temporal distribution of the study's data is uneven due to the availability of data and the actual development of the study area. This research revealed changes in land use and landscape patterns in the study area over seven periods, forming six stages. Although there are differences in the period of these six time periods, they still clearly reflect the dynamic changes in land use and landscape patterns in the study area. Secondly, the scope of the study area did not coincide with the scope of the township, and precise socioeconomic data were difficult to obtain, thus a detailed quantitative analysis of the driving mechanisms was lacking. Further studies will investigate the driving mechanisms and effects of the transition of IBPFU in karst mountain areas to provide a more reliable reference base for ensuring food security in karst mountain areas.

**Author Contributions:** Conceptualization, M.C. and Y.L.; methodology, M.C.; software, M.C. and F.T.; validation, M.C.; formal analysis, M.C.; investigation, M.C., Y.L., F.T., Q.X., M.Y. and H.Z.; resources, M.C., Y.L., F.T., Q.X., M.Y., H.Z. and X.L.; data curation, M.C.; writing—original draft preparation, M.C.; writing—review and editing, M.C., Y.L., F.T., Q.X., M.Y., H.Z. and X.L.; visualization, M.C., Y.L., F.T., Q.X., M.Y., H.Z. and X.L.; supervision, M.C., Y.L., F.T., Q.X., M.Y., H.Z. and X.L.; project administration, M.C. and Y.L.; funding acquisition, Y.L. All authors have read and agreed to the published version of the manuscript.

**Funding:** This research was funded by the National Natural Science Foundation of China, grant number 41661020 and 4261035; by Guizhou Science and Technology Cooperation Platform Talents [2021] A22 and [2017]5726-54.

**Data Availability Statement:** Not applicable.

**Conflicts of Interest:** The authors declare no conflict of interest. The funders had no role in the study; in the collection, analyses, or interpretation of data; in the writing of the manuscript, or in the decision to publish the results.

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
