# Peer review of "Transformation of Paddy Field Use in Intermountain-Type Basins Using Evidence from the Structure and Function Perspective of Karst Mountain Areas in Southwest China"

_agronomy, doi:10.3390/agronomy13061552_

Round 1

Reviewer 1 Report

The Authors of the Article “Transformation of paddy field use in the intermountain basin from the structure and function perspective-evidence of karst mountain areas in Southwest China“ analyzed the evolution and transformation patterns of intermountain basins paddy fields use (IBPFU) to optimize land resources in mountainous area. Based on the structural and functional perspectives, this study established a temporal-quantitative-spatial coupling research framework to analyze the evolution and transformation patterns of typical IBPFU in Guizhou Province from 1966 to 2020.

The article is in line with current research trends and can be taken as an extension of knowledge in the analyzed topic.

The article is properly developed, containing all the elements that should be included in a good scientific article.

In my opinion, in order to improve the quality of the article, minor changes are required in it, mainly concerning the ordering of the methodology. First of all, I miss a clear indication of what new contributions this study makes to knowledge in the topic under analysis. This needs to be clearly highlighted in the article.

In the current version of the article, only elements related to the inventory of land use changes, their dynamics and spatial patterns are methodically described. In terms of methodology, were the authors concerned only with such elements? I believe that much more valuable are the elements contained, for example, in Fig.1 (described in the methodology only in six lines 127-131). Are the elements shown in Fig.1 a proposal of the Authors or were they developed on the basis of other publications (the footnote in line 127 raises doubts in this regard)? If it is not the Authors' proposal of the article, why is it in this chapter?

I have similar doubts about Figure 12 from the Discussion section. Is this a proposal by the Authors, or was this diagram developed from other research - no source given next to the figure. If this diagram was developed by the Authors then it is more in line with the results of the research, only it would have to be announced methodically beforehand. If, on the other hand, the diagram presents other research, why is it here? Do the authors refer to it with their results? Please put it in order.

In my opinion, the methodology of the article needs some structuring. I propose to add a diagram to the article showing the research process of this article, and in the next steps indicate the research methods used. All of this must, of course, be consistent with the stated purposes of the study.

If the elements in Figures 1 and 12 are the Authors' proposal then the whole article just needs some rewriting and supplementation. If, on the other hand, these elements are not Authors', then some problems will arise.  The article then will basically be just an inventory of changes, and the given conclusions and elements from the discussion of the results will not fully follow from the research, they will only be the subjective opinion of the Authors.

Author Response

Dear Reviewers,

Thank you for taking the time out of your busy schedule to read and modify my manuscript (Manuscript Number: Agronomy-2413682). You have corrected all aspects of the methodology, highlight, figure, and content of my paper, which will play a significant role in improving the quality of my paper. We have carefully revised the manuscript based on your suggestions and uploaded the specific revision instructions to our system. Please see the attachment "response of author." Thank you again for your kind attention to our manuscript.

Wish you good work and happy life!

Best regards, 

All Authors

Reviewer 2 Report

Review on Transformation of paddy field use in the intermountain basin (please review the title of the paper, which basin are you referring to?) from the structure and function perspective-evidence of karts mountain (mountain or intermountain??) areas in Southwestern China.

Rows 15-16…I believe that the typical landforms in karst area include, beside intermountain basins, some ridges that delineate them.

Row 21 . Please define the difference between reversibility an irreversibility, in the context given.

Rows 30-31 – Where come in this affirmation the agricultural terrains of USA or Ukraine, not mentioning other countries, in what regards global food security?

Starting from above, but mentioning only rows 33-34….a serious need of English language is required.

Row 45 – please define how the driving mechanisms of the change mentioned are less studied?

Row 57….serious revision of English, so as rows 61-62, 84-85…some phrases can be interpreted in other ways. I will stop here from mentions about the use of English.

Row 45-46 – the affirmation is quite doubtful, taking into consideration the long list of references

References have been quite clearly covered.

Row 100…in contrast to the slopes …vs IBs…maybe a different landform classification could be used? Slopes go from 1 degree…..so I think that IBs also have slopes.

Please use either mu or sqm / sqkm as measurement. Or better SI.

144-145 – Also, use an international system of soil classification, such as WRB.

Lines 153-155 – is this change more favourable for the inhabitants?

Nice array of data used. Still, the image attached to table 2 might need a better resolution.

Review on Transformation of paddy field use in the intermountain basin (please review the title of the paper, which basin are you referring to?) from the structure and function perspective-evidence of karts mountain (mountain or intermountain??) areas in Southwestern China.

Rows 15-16…I believe that the typical landforms in karst area include, beside intermountain basins, some ridges that delineate them.

Row 21 . Please define the difference between reversibility an irreversibility, in the context given.

Rows 30-31 – Where come in this affirmation the agricultural terrains of USA or Ukraine, not mentioning other countries, in what regards global food security?

Starting from above, but mentioning only rows 33-34….a serious need of English language is required.

Row 45 – please define how the driving mechanisms of the change mentioned are less studied?

Row 57….serious revision of English, so as rows 61-62, 84-85…some phrases can be interpreted in other ways. I will stop here from mentions about the use of English.

Row 45-46 – the affirmation is quite doubtful, taking into consideration the long list of references

References have been quite clearly covered.

Row 100…in contrast to the slopes …vs IBs…maybe a different landform classification could be used? Slopes go from 1 degree…..so I think that IBs also have slopes.

Please use either mu or sqm / sqkm as measurement. Or better SI.

144-145 – Also, use an international system of soil classification, such as WRB.

Lines 153-155 – is this change more favourable for the inhabitants?

Nice array of data used. Still, the image attached to table 2 might need a better resolution.

Author Response

Dear Reviewers,

Thank you for taking the time out of your busy schedule to read and modify my manuscript (Manuscript Number: Agronomy-2413682). You have corrected all aspects of the title, abstract, introduction, content, language, and results of my paper, which will play a significant role in improving the quality of my paper. We have carefully revised the manuscript based on your suggestions and uploaded the specific revision instructions to our system. Please see the attachment "response of author." Thank you again for your kind attention to our manuscript.

Wish you good work and happy life!

Best regards, 

All Authors

Reviewer 3 Report

1. Please specify the method in the abstract, quantitative-spatial coupling research framework is not well explained.

2. Please mention the objective in the abstract clearly.

3. On Figure 2, the authors compare the field in 2014 and 2022. Please explain the history more detail. Why did paddy field become economic forest?

4. What kind of paddy variety in the study area?

5. I wonder the evolutionary pattern of paddy field use in the Figure 11. How can these happen in the karst? why farming can be submerged in water like in the picture. Though karst land has the characteristics of lack of water.

6. Is there no evolution to be housing area?

7. Please complete each finding with empirical evidences

8. Is there any livestock in the study area?

Author Response

Dear Reviewers,

Thank you for taking the time out of your busy schedule to read and modify my manuscript (Manuscript Number: Agronomy-2413682). ou have corrected all aspects of the abstract, introduction, content, figure, and results of my paper, which will play a significant role in improving the quality of my paper. We have carefully revised the manuscript based on your suggestions and uploaded the specific revision instructions to our system. Please see the attachment "response of author." Thank you again for your kind attention to our manuscript.

Wish you good work and happy life!

Best regards, 

All Authors
